# Spelling Correction for Estonian Learner Language

**Kais Allkivi-Metsoja**
School of Digital Technologies
Tallinn University
`kais@tlu.ee`

**Jaagup Kippar**
School of Digital Technologies
Tallinn University
`jaagup@tlu.ee`

## Abstract

Second and foreign language (L2) learners tend to make specific spelling errors compared to native speakers. Language-independent spell-checking algorithms that rely on n-gram models can offer a simple solution for improving learner error detection and correction due to context-sensitivity. As the open-source speller previously available for Estonian is rule-based, our aim was to evaluate the performance of bi- and trigram-based statistical spelling correctors on an error-tagged set of A2–C1-level texts written by L2 learners of Estonian. The newly trained spell-checking models were compared to existing correction tools (open-source and commercial). Then, the best-performing Jamspell corrector was trained on various datasets to analyse their effect on the correction results.

## 1 Introduction

It has been proposed that tailor-made spelling error correction systems are best suited for language learning purposes because the spell-checking tools developed for proficient users often prove unable to correct specific mistakes, like real-word errors, i.e., errors that result in a valid homonym; diacritic errors; or pronunciation-induced errors possibly with a large edit distance (e.g., Lawley 2016). Whereas it is costly to develop rule-based error correction systems with learner-oriented explanations, or neural spell-checking systems that require vast quantities of training data comprising authentic or synthetic errors, statistical spelling correction algorithms which use n-gram language models to analyse context could form a simple starting point for improving error detection and correction of L2 learner writings. In this language-independent approach, only a corpus of (presumably) correct language use samples is needed to train the system.

Currently, the only open-source spell-checker developed for Estonian language is Vabamorf[1]. It is a lexicon- and rule-based library created by Filosoft Ltd. at the beginning of the 1990-s alongside a commercial speller distributed in Microsoft Word (Kaalep et al., 2022). The spellers make use of a lexicon and a list of typing misspellings to assess candidate corrections but they do not appear to rely on context in their suggestions.

For evaluating statistical spelling error detection and correction on Estonian learner language, we first used Peter Norvig's approach that generates all possible spelling corrections by different edits, such as character deletions, insertions, replacements, and transpositions (Norvig, 2007). The procedure is repeated to get correction candidates with two edits. The probability of candidates is estimated based on a unigram language model derived from a language corpus. We used a bigram language model in addition to a unigram model to add context-sensitivity.

Second, we applied the compound aware version of Symmetric Delete Spelling Correction (Symspell)[2]. The algorithm searches for candidate corrections with an edit distance of 1 or 2 based on deletions only, increasing the speed of spelling correction. A corpus-based bigram dictionary can be used, however, bigrams are only considered in ranking suggestions if no suggestions with an edit distance of 1 are found for a single token. Real-word spelling errors are currently not corrected by Symspell (Garbe, 2017).

Third, we tested Jamspell[3] that additionally uses a trigram language model for selecting the highest-scored correction candidate. Jamspell is

---

[1] `https://github.com/Filosoft/vabamorf`
[2] `https://github.com/wolfgarbe/SymSpell`
[3] `https://github.com/bakwc/JamSpell`

based on a modified Symspell algorithm, optimized for speed and memory usage, so that the spell-checking library can process up to 5,000 words per second.

We compared the algorithms with three existing spell-checking tools: Vabamorf, and the commercial spellers offered by MS Word (Microsoft 365) and the Google Docs application. The latter uses neural machine translation (Kumar and Tong, 2019).

## 2 Test Data and Evaluation

The correction output was evaluated on a set of 84 error-annotated proficiency examination writings from the Estonian Interlanguage Corpus[4]. Divided between four proficiency levels (A2, B1, B2, and C1), the texts contain 1,054 sentences, 9,186 words (excluding anonymized identifiers), and 309 spelling errors in total. We distinguished simple spelling errors and mixed errors, i.e., spelling mistakes co-occurring with another error such as word choice, inflectional form, or capitalization error. The error distribution is given in table 1. While the proficiency level increases, the relative frequency of words containing a spelling error decreases, from 5.5% at A2 and 3.8% at B1 to 2.6% at B2 and 2.3% at C1.

| Proficiency level | Words | Simple spelling errors | Mixed errors |
|---|---|---|---|
| A2 | 1,852 | 73 | 28 |
| B1 | 2,186 | 71 | 12 |
| B2 | 2,074 | 51 | 3 |
| C1 | 3,074 | 68 | 3 |
| Total | 9,186 | 263 | 46 |

Table 1: Spelling correction test data.

The texts have been morphologically annotated in the CoNLL-U format[5], using the Stanza toolkit[6], and manually error-tagged, indicating the error type, scope, and correction in the field for miscellaneous token attributes. While the custom tagset denotes various orthographic and grammatical errors, we only rated the detection and correction of words annotated to have a spelling error (although, we did not count a system edit as

unnecessary if the word had any error tag). Each text has been reviewed by two annotators, consulting a third Estonian language expert in case of disagreement. The annotation format allows for several corrections per token but is limited to one error annotation per sentence. This, however, has no significant effect on the analysis of spelling errors, which occur regardless of the sentence structure.

Error detection is the first step of error correction. Nevertheless, to achieve high performance in error detection, the proposed edits do not have to match the gold standard annotation, as opposed to measuring error correction performance. We evaluated both spelling error detection and correction based on three metrics:

- **recall** – the percent of spelling errors detected/corrected;

- **precision** – the percent of relevant/correct changes made;

- **F0.5 score** – a combined measure of precision and recall that gives precision twice as much weight as recall.

The F0.5 score was preferred to the harmonic mean (F1 score) due to the assumption that an error correction system's reliability is rather reduced by false and needless corrections than unproposed corrections (see Ng et al. 2014).

We verticalized the system output and automatically compared it to the test set to detect changes and correction matches. Since L2 learners may not select the correct option from a list of suggestions (e.g., Heift and Rimrott 2008) and such selection cannot be implemented in an automated workflow, e.g., using spell-checking as a pre-processing step of grammatical error correction, we prioritized the speller's accuracy of defining the best correction. Thus, we focused on the highest-ranked suggestion. The cases of mixed errors were reviewed manually to find partial corrections fixing only the spelling of an otherwise erroneous word (e.g., *parnu*~*pärnu* instead of *Pärnu*, which is an Estonian town name and should be capitalized). Both full and partial word corrections were considered in calculating the evaluation metrics.

## 3 Comparison of Spell-Checking Tools

The training material for building new statistical spell-checking models came from the Estonian National Corpus (ENC) 2019, which includes web

---

[4]https://evkk.tlu.ee/about/us/
[5]https://universaldependencies.org/format.html/
[6]https://stanfordnlp.github.io/stanza/

corpora downloaded from Estonian websites as well as the Estonian Reference Corpus, Wikipedia corpora and the corpus of Estonian Open Access Journals (DOAJ) (Koppel and Kallas, 2020). Jamspell and Norvig's spelling corrector were trained on a random sample of 6 million sentences and over 82 million words retrieved from the Reference Corpus that represents the "standard" varieties of Estonian – mostly newspaper texts but also fiction, science and legislation texts from 1990–2008. The sample constitutes nearly half of the Reference Corpus; increasing the volume of the training set did not improve the correction results. Symspell, on the other hand, reached the best results with a uni- and bigram frequency dictionary based on the full ENC 2019 containing over 1.5 billion words. Even then, it performed poorly compared to other tools, especially in terms of recall.

The comparison of spelling error detection and correction by the different applications is summarized in tables 2 and 3. Table 4 shows the distribution of system edits which can be relevant, resulting in identified errors, or unnecessary, leading to broken words. Relevant edits that do not match the expert correction are considered false corrections.[7]

Jamspell and Norvig's speller outperformed Vabamorf and Word's speller in error correction, and Google's spell-checker in error detection. All in all, Google corrected the highest proportion of spelling errors, followed by Jamspell, which still had a significantly better correction recall than the rest of the tools and came close to Google in terms of correction precision and F0.5 score. Despite a larger number of accurate corrections, Google made more than twice as many unnecessary edits.

| Spell-checker | F0.5 | Precision | Recall |
|---|---|---|---|
| Jamspell | 83.9 | 89.6 | 67.0 |
| Norvig | 78.9 | 84.3 | 62.8 |
| Symspell | 69.1 | 86.2 | 38.5 |
| Google | 76.7 | 78.8 | 69.6 |
| MS Word | 83.4 | 87.8 | 69.6 |
| Vabamorf | 84.3 | 89.2 | 69.3 |

Table 2: Spelling error detection metrics (%).

[7]The correction outputs as well as the test material can be found at https://github.com/tlu-dt-nlp/spell-testing/.

| Spell-checker | F0.5 | Precision | Recall |
|---|---|---|---|
| Jamspell | 64.1 | 68.4 | 51.1 |
| Norvig | 54.1 | 57.8 | 43.0 |
| Symspell | 31.4 | 39.1 | 17.5 |
| Google | 67.5 | 69.2 | 61.2 |
| MS Word | 51.2 | 53.9 | 42.7 |
| Vabamorf | 42.6 | 45.0 | 35.0 |

Table 3: Spelling error correction metrics (%).

In error detection, Jamspell yielded results similar to Vabamorf and MS Word. Norvig's spell-checker and Symspell also scored better than Google in detection precision. While Symspell broke the smallest number of words at the cost of very low recall, the lowest percent of unnecessary edits was achieved by Jamspell and Vabamorf – 10.4% and 10.8% respectively. At the same time, 21.2% of words edited by Google did not need to be corrected.

If matching candidate suggestions were considered, the spell-checking tools would reach a higher correction precision, except for Google's speller that proposed only a single correction. Vabamorf's precision (72.5%) would increase the most, Jamspell's precision (72.3%) the least. It means that Jamspell is more likely to suggest an accurate correction with the highest confidence.

Compared to their open-source counterpart Vabamorf, both Jamspell and Norvig's speller benefit from relying on the context of erroneous words. For example, Vabamorf corrected the verbs *tõdida~tõdeda* 'admit-INF' and *ludeda~lugeda* 'read-INF' to *tüdida* 'get.bored-INF' and *kudeda* 'spawn-INF'. Interestingly, the rule-based spell-checker tended to replace other parts-of-speech with nouns, e.g., the adverb *lahtii~lahti* 'open' was changed to *Lahti*, a location in Finland, and the adverb *nanuke~natuke* 'a bit' to *januke* 'thirst-DIM'. Real-word spelling errors inducing homonymy were best handled by Jamspell that was able to make corrections such as *vaga~väga* 'very' (*vaga* could be an adjective meaning 'pious, godly'); *töökohtu~töökohti* 'job-PART.PL' (*töökohtu* could mean 'labour.court-GEN.PL'); and *kuued~kuud* 'month-PART.SG' (*kuued* could be a numeral meaning 'six-NOM.PL' or a noun meaning 'coat-NOM.PL').

Like Google's spell-checker, Jamspell and Norvig's speller occasionally attempted to cor-

| Spell checker | Errors detected | Full corrections | Partial corrections | Broken words |
|---|---|---|---|---|
| Jamspell | 207 | 129 | 29 | 24 |
| Norvig | 194 | 108 | 25 | 36 |
| Symspell | 119 | 45 | 9 | 19 |
| Google | 215 | 163 | 26 | 58 |
| MS Word | 215 | 108 | 24 | 30 |
| Vabamorf | 214 | 88 | 20 | 26 |

Table 4: Changes made by spell-checkers.

rect word choice and inflectional form, although merely a couple of mixed errors were fully corrected (e.g., *seles*~*sellel laupäeval* 'this Saturday' where the misspelled inessive pronoun was replaced with the correctly spelled adessive form agreeing with the noun). Otherwise, we only took such edits into account if they were unnecessary and resulted in a broken word. It can, however, be noted that Jamspell was more probable to make accurate lexical and grammatical corrections than Norvig's corrector, given a small edit distance, e.g.,*ennem* 'rather'~*enne* 'before', *kümne* 'ten-GEN.SG'~*kümme* 'ten.NOM.SG'. Similarly to Vabamorf and MS Word, Norvig's speller replaced some proper nouns with common nouns, e.g., *Kemeris* 'Kemer-IN.SG' referring to a Turkish location was corrected to *Keeris* 'vortex.NOM.SG'. Such behaviour was the most characteristic to Vabamorf which also proposed changes to rather common first names, e.g., *Nadja*~*Andja* 'giver'. Furthermore, some unnecessary edits made by Google, Word and Symspell were caused by splitting compound words.

On the other hand, it should be noted that the statistical spell-checkers do not correct capitalization because all words are transformed to lowercase when processing the text and then printed in the original casing. In general, all the tested spelling correction tools struggled with proposing the right correction instead of a candidate with a smaller edit distance (e.g., *musi-ika*~*muusika* 'music' was corrected as *mustika* 'blueberry.GEN.SG'; *sõidata*~*sõita* 'ride-INF' as *sõimata* 'curse-INF').

In conclusion, two of the tested statistical spell-checkers achieved a better precision and recall in correcting Estonian L2 learners' spelling errors compared to the existing open-source speller Vabamorf . Jamspell's performance was similar to

MS Word in error detection and comparable with Google in error correction, the main difference being that Google corrected more spelling errors at the cost of making more unnecessary edits. Therefore, Jamspell should be favoured if the priority is to minimize needless corrections.

## 4 Jamspell Correction Models

We experimented with different training data to see if we can improve Jamspell's efficiency in learner spelling error detection and correction. The training sets are listed in table 5.

| Training corpus | Sentences | Words |
|---|---|---|
| Web 2019 | 40,880,346 | 512,567,596 |
| Reference + Wikipedia + DOAJ | 16,935,524 | 230,066,343 |
| Reference | 13,173,122 | 180,944,778 |
| Web 2019 sample | 6,000,000 | 75,237,791 |
| Reference sample | 6,000,000 | 82,401,187 |
| Reference + Web 1:1 | 6,000,000 | 78,855,570 |
| Reference + Web 10:1 | 6,600,000 | 89,921,477 |
| Reference + Wikipedia + DOAJ sample | 4,172,777 | 55,743,160 |

Table 5: Data for training Jamspell models.

On the one hand, we combined the Estonian Reference Corpus with the DOAJ and Wikipedia corpora of ENC 2019. These subcorpora contain, to a large extent, language-edited texts. As the Reference Corpus constitutes the majority of this

training set, we also extracted a more balanced sample, which includes an equal amount of randomly chosen sentences from the Reference Corpus and Wikipedia corpora as well as the whole DOAJ corpus (442,663 sentences). On the other hand, we trained Jamspell on the Estonian Web corpus 2019 that comprises a more diverse selection of texts, from informal blog posts and forum discussions to periodicals and educational materials. We used the full corpus and a sample similar in size with the Reference Corpus sample. Thirdly, we merged the Reference Corpus and Web 2019 material in an equal ratio and in a ratio of 10:1, giving emphasis to the more "standardized" texts and using the web texts to add variation to the dataset.

The results of spelling error detection and correction obtained on the previously used test set are presented in tables 6 and 7. The system edit distribution is provided in table 8. In case of similar training datasets (full corpus and sample), the lower-performing correction model has been omitted. Models trained on samples of the Reference Corpus and its combination with other edited subcorpora achieved better or similar results compared to the models trained on full text sets. Contrary to that, the model trained on the whole Estonian Web 2019 performed better than the model based on the web sample in all aspects.

The comparison of the Jamspell models reflects the well-known trade-off between precision and recall. The highest error detection and correction precision were achieved by the model trained on Estonian Web 2019. It was the least likely to make unnecessary corrections but also to detect words with a spelling error, thus having the lowest recall. At the same time, the initial model trained on a Reference Corpus sample scored highest in error detection and correction recall, being able to identify and correct the largest amount of spelling errors. The latter model featured the best F0.5 score in error detection, whereas the Web 2019 model had a slightly better F0.5 score in error correction.

In terms of spelling error detection, the 10:1 Reference + Web sample offered a compromise, yielding a higher precision than the Reference Corpus model and a higher recall than the Estonian Web model. This resulted in the second best F0.5 score. On the other hand, there was little variation in the error correction F0.5 score. The performance obtained with the 10:1 Reference + Web

| Training corpus | F0.5 | Precision | Recall |
|---|---|---|---|
| Reference sample | **83.9** | 89.6 | **67.0** |
| Reference + Web 10:1 | 82.7 | 91.2 | 60.2 |
| Web 2019 | 81.9 | **94.3** | 53.7 |
| Reference + Wikipedia + DOAJ sample | 80.4 | 87.7 | 60.2 |
| Reference + Web 1:1 | 79.9 | 89.6 | 55.7 |

Table 6: Spelling error detection metrics of Jamspell models (%), ranked by F0.5 score.

| Training corpus | F0.5 | Precision | Recall |
|---|---|---|---|
| Web 2019 | **64.7** | **74.4** | 42.4 |
| Reference sample | 64.1 | 68.4 | **51.1** |
| Reference + Wikipedia + DOAJ sample | 63.5 | 69.3 | 47.6 |
| Reference + Web 10:1 | 63.1 | 69.6 | 46.0 |
| Reference + Web 1:1 | 63.1 | 70.8 | 44.0 |

Table 7: Spelling error correction metrics of Jamspell models (%), ranked by F0.5 score.

sample was almost identical to the model trained on the Reference + Wikipedia + DOAJ sample. The 1:1 Reference + Web sample model scored slightly higher in correction precision and lower in correction recall.

Concerning the relation between the training corpus type and size, and the performance of the spell-checking model, we may infer that a smaller, "standard language" dataset rather facilitates higher recall. Increasing the dataset introduces more noise, thus the errors are outlined less clearly. A much larger and more diverse language model leads to higher precision; decreasing the dataset reduces lexical variation and entails more unnecessary edits. For comparison, the Web 2019 trigram model consists of 279.1 million trigrams, whereas the model trained on the Reference Corpus sample has 52.8 million trigrams.

The choice of the most suitable model depends

| Training corpus | Errors detected | Full corrections | Partial corrections | Broken words |
|---|---|---|---|---|
| Reference sample | **207** | **129** | **29** | 24 |
| Reference + Wikipedia + DOAJ sample | 186 | 122 | 25 | 26 |
| Reference + Web 10:1 | 186 | 116 | 26 | 18 |
| Reference + Web 1:1 | 172 | 113 | 23 | 20 |
| Web 2019 | 166 | 115 | 16 | **10** |

Table 8: Jamspell models ranked by spelling errors detected and corrected.

on the purpose – whether we want to maximize the amount of errors detected and corrected, minimize the amount of needless corrections, or find a middle ground. For this, combining a larger proportion of standard texts with a smaller proportion of web material seems the best suited. In summary, the results are promising compared to the precision and recall of learner spelling error correction accomplished in other languages (e.g., Bexte et al. 2022; Kantor et al. 2019).

Three best-performing Jamspell models have been made available for use as a part of the new Estonian spelling and grammatical error correction toolkit currently in development[8].

## 5 Conclusion and Perspectives

This study has demonstrated the benefit of statistical context-sensitive spelling correction for processing L2 learner writings. Jamspell that uses trigram contexts of words for spell-checking could correct real-word errors and other learner-specific spelling errors more efficiently than other tested open-source spellers. In spelling error correction, it also outperformed MS Word speller, achieving precision and recall comparable to Google's corrector. In spelling error detection, its performance was similar to MS Word's and better than Google's. The evaluation of different Jamspell correction models revealed that using a web corpus as training material increases error detection and correction precision, while using a reference corpus increases recall.

We consider the current correction models a decent baseline for further development. Their performance could be improved, e.g., by employing learner spelling error frequency data or named-entity recognition to avoid false name edits and

enable correction of name capitalization.

We acknowledge that the results might have been different if we had implemented Norvig's spell-checking algorithm on a trigram language model. The tested spell-checking tools and models should also be evaluated on a larger error-annotated set of writings by L2 learners as well as native speakers. Such a gold-standard dataset of approximately 8,000 sentences is in development for Estonian. Expectedly, context-sensitive spelling correction also benefits proficient language users, although the difference in performance may not be as outstanding.

## Acknowledgments

This research has been funded by the national programme "Estonian Language Technology 2018-2027" and the Tallinn University Research Fund. We thank Marko Kollo, Rico-Andreas Lepp and Martin Mõtus for their help in testing the spell-checking tools. We also thank Kaisa Norak, Linda Luig and Pille Eslon for working on error annotation.

---

[8]The repository of the collaborative project with the University of Tartu can be accessed at https://koodivaramu.eesti.ee/tartunlp/corrector/-/tree/main/.

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
