# OpenReview forum: "Spelling Correction for Estonian Learner Language"
_NoDaLiDa/2023/Conference — NoDaLiDa 2023_

### Official Review · Reviewer_Nybe · 2023-02-20
**Valuable evaluation of Estonian spelling correction methods and systems, with some fixing needed**

**Rating:** 7
**Confidence:** 4

**Review:**

This paper evaluates a number of existing spelling correction systems for Estonian, as well as exploring training data for an n-gram based context sensitive spell checker. The evaluation is well-designed and detailed, using a collection of corrected learner texts at different proficiency levels. This alone is sufficient reason to accept the paper, I think.

Some clarifications and fixes are needed:

- bigram and trigram language models are not really the state of the art, but I recognize that they can still be useful in computationally constrained or low-resource scenarios. The use of such a simple model should be justified, and perhaps even supported by performance figures (words per second, RAM usage, etc., which is already mentioned for Jamspell but not the others).
- a summary of which types of methods are used by the different spell checkers would be valuable, to the extent that this information is public for the commercial systems. For instance, are any of the systems based on neural language models?
- does a "full" correction of mixed errors (as opposed to partial) mean that the other error was also fixed? For instance, changing to a correct inflection. Some further detail and/or examples here would be helpful to make the reader understand the distinction between the categories "full" and "partial" used in the evaluation, and to what extent some of the systems are also performing more general grammatical error correction.
- the reference list is missing. The in-text citations look OK, so I assume there was some error while producing the manuscript.

**Paper Type:**

Short paper

---

### Official Review · Reviewer_mwmc · 2023-03-03
**The paper would need considerable re-writing in order to be published. The evaluation data would require major changes in the prose of the article also. I would recommend the authors revise their paper and re-submit it somewhere else; their work has lots of merit and it deserves to be published.**

**Rating:** 4
**Confidence:** 4

**Review:**


Strong points of the paper

The paper is about an interesting question: what would be the best approach and/or tool for correcting texts written by learners of Estonian as a second language. The paper describes how a machine learning tool Jamspell was trained for Estonian, and compares its performance with other ML approaches (which the authors have trained themselves) and rule-based approaches. Training all these tools must have taken considerable effort, being a valuable exercise, although it is a pity that the resulting data files are not made public (at least the paper does not mention them explicitly). The test corpus is one of real life errors, which is really good, although this has not been made public either.

Weak points of the paper

The paper has the following deficiencies:
1. The bibliography is missing. In other words, the paper does not formally qualify as a submission.
2. Evaluation and ranking of the spell checkers has conceptual flaws, as explained below in detail.
3. Explanation of the calculations is hard to follow. In fact, I had to create my own spreadsheet, fill it with numbers and try out various formulas in order to understand what the figures in tables 2 and 3 actually mean.

Conceptual flaws

One reason for difficulties in evaulation and ranking is that the problem of finding and correcting errors is approached differently in traditional rule-based and machine learning approaches. The traditional approach assumes that there are two separate tasks: first, identify an erroneous word, and after that, suggest some correction (usually present a list of possible ones). The machine learning approach assumes that there is one task: suggest another word to replace the one that seems to be unfit for the context. So the ML approach has no problem to always suggest exactly one word; the problem is to decide whether the original word is "bad enough" for needing replacement. However, when comparing the traditional rule-based and ML approaches, one still has to conceptualize the problem as consisting of two separate tasks similar to the traditional view - this will give a better understanding of the strengths of the approaches.

The authors seem to try to find one aggregate metric, conflating the error detection and correction suggestion tasks. Their solution, however, looks like an ad hoc measure introduced just to show that certain tools are ranked better; other measures would give a different ranking.

Usually, precision and recall are defined via a confusion matrix containing figures for true positives (TP), true negatives (TN), false positives (FP) and false negatives (FN), by the following formulae: recall = TP/(TP+FN);  precision = TP/(TP+FP).

The figures TP, TN, FP and FN are counted on a test set, and it is important they are counted on the SAME test set. This is a trivial truth, but unfortunately the authors have ignored this. Recall and precision are calculated on different test sets (page 2). Table 3 presents figures that have been calculated in the following way:

Table 3: Recall = (Table 2: Errors detected) / (True errors)

Table 3: Precision = (Table 2: Full corrections + Partial corrections) / (Table 2: Errors detected + Broken words)

Table 3: Errors corrected = (Table 2: Full corrections + Partial corrections) / (True errors)

The number of true errors is 309 (Table 1).

The combined measure of F0.5 is hard to interpret because it combines weighted results from different test sets.

Calculating Recall and Precision separately on error detection and correction would result in the following two tables, indicating that in error detection, Vabamorf and Jamspell are the best; in error correction, Google is the best. The ranking of tools would be somewhat different from the one presented in the paper. However, Jamspell would be near the top, so the conclusion of the paper that Jamspell is a good tool, is justified.

Recall and precision of error DETECTION.

Recall = Table 3: Recall = (Table 2: Errors detected) / (True errors)

Precision = (Table 2: Errors detected) / (Table 2: Errors detected + Broken words)

|         | Recall | Precision
| ----    | -----  | -----
|Jamspell | 67,0% | 89,6%
|Norvig   | 62,8% | 84,3%
|Symspell | 38,5% | 86,2%
|Google   | 69,6% | 78,8%
|MS Word  | 69,6% | 87,8%
|Vabamorf | 69,3% | 89,2%

The article lacks this column of Precision, although it contains a remark on p. 3 "The lowest percent of unnecessary edits among all ..." citing figures for (1 - precision) for Jamspell and Vabamorf.

Recall and precision of error CORRECTION, assuming that only the first suggested correction is taken into consideration. (If a tool proposes a suitable correction, but it is not presented as the first one in a list of several possible suggestions, then this correction is regarded as if not presented at all. This disfavors the traditional tools.)

Recall = (Table 2: Full corrections + Partial corrections) / (Table 2: Errors detected)

Precision = Table 3: Precision = (Table 2: Full corrections + Partial corrections) / (Table 2: Errors detected + Broken words)

|         | Recall | Precision
| ----    | -----  | -----
|Jamspell | 76,3% | 68,4%
|Norvig   | 68,6% | 57,8%
|Symspell | 45,4% | 39,1%
|Google   | 87,9% | 69,2%
|MS Word  | 61,4% | 53,9%
|Vabamorf | 50,5% | 45,0%

The article lacks this column of Recall.

Conclusion

The paper would need considerable re-writing in order to be published. The evaluation data would require major changes in the prose of the article also. I would recommend the authors revise their paper and re-submit it somewhere else; their work has a lot of merit and it deserves to be published.




**Paper Type:**

Long paper

---

### Official Review · Reviewer_Byg1 · 2023-03-10
**Comparison paper; interesting results**

**Rating:** 9
**Confidence:** 4

**Review:**

The paper compares 6 existing spell-checkers applied to an Estonian L2 learner corpus. Three of the tools are based on edit-distance and n-grams and can thus be trained on the available Estonian corpus. The results are interesting (but perhaps not surprising), showing the effectiveness and simplicity of n-gram-based methods and that one can easily reach Google-level quality with open-source solutions (for Estonian). The paper is easy to read and answers most questions. More discussion could be dedicated to the differences between L1 and L2 errors, i.e., do the authors imagine that the results would be any different if the tools were to be applied to an L1 corpus. Also, it remained unclear why the recall peaks at 70%. It would be nice to have more examples of spelling errors that cannot be easily detected, and the reasons for it (long-distance dependences, OOVs, differences with edit-distance > 2?). Do the authors imagine that more modern NLP techniques (word vectors, attention, word pieces) would detect these errors?

Remark:

- Section 1 claims that "Vabamorf is rule-based, which means that is does not rely on context". This is misleading. While Vabamorf (perhaps?) analyses words in isolation, rule-based methods, in general, can often "access" more context (e.g., via context-free grammar rules) than n-gram methods.

**Paper Type:**

Long paper

---

### Decision · Program_Chairs · 2023-03-17

Accept